# Helminth Lessons in Inflammatory Bowel Diseases (IBD)

**DOI:** 10.3390/biomedicines11041200

**Published:** 2023-04-18

**Authors:** Tyler Atagozli, David E. Elliott, Mirac Nedim Ince

**Affiliations:** 1Department of Internal Medicine, Division of Gastroenterology and Hepatology, University of Iowa Carver College of Medicine, Iowa City, IA 52246, USA; 2Iowa City Veterans Affairs Medical Center, Iowa City, IA 52246, USA

**Keywords:** ulcerative colitis, Crohn’s disease, inflammatory bowel diseases (IBD), helminths, immune regulation

## Abstract

Helminths are multicellular invertebrates that colonize the gut of many vertebrate animals including humans. This colonization can result in pathology, which requires treatment. It can also lead to a commensal and possibly even a symbiotic relationship where the helminth and the host benefit from each other’s presence. Epidemiological data have linked helminth exposure to protection from immune disorders that include a wide range of diseases, such as allergies, autoimmune illnesses, and idiopathic inflammatory disorders of the gut, which are grouped as inflammatory bowel diseases (IBD). Treatment of moderate to severe IBD involves the use of immune modulators and biologics, which can cause life-threatening complications. In this setting, their safety profile makes helminths or helminth products attractive as novel therapeutic approaches to treat IBD or other immune disorders. Helminths stimulate T helper-2 (Th2) and immune regulatory pathways, which are targeted in IBD treatment. Epidemiological explorations, basic science studies, and clinical research on helminths can lead to the development of safe, potent, and novel therapeutic approaches to prevent or treat IBD in addition to other immune disorders.

## 1. Helminths as Infectious or Commensal Agents

Helminths are small invertebrate organisms, many of which are visible to the naked eye. They are divided into two phyla, nematodes (round worms) and platyhelminths (flat worms) [1]. Platyhelminths are further divided into two classes, cestodes (tapeworms), and trematodes (flukes). Although all are called worms, nematodes, cestodes, and trematodes are phylogenetically very distant and parasitism developed independently—even between members of the same class [2]. Each helminth species has successfully developed strategies to infect a host, replicate, and spread to additional hosts. This lifecycle can be direct person-to-person, e.g., *Enterobius vermicularis* (pinworm), or indirect through intermediate hosts, e.g., *Schistosoma mansoni* (schistosomiasis). Many of the most pathogenic helminths infect humans inadvertently (accidental hosts), e.g., *Capillaria philippinensis*. Their specific lifecycle defines their geographic range, the method for acquiring infection, and which host tissues are most affected.

Helminth infections are most prevalent in geographies where sanitary conditions are poor and are often endemic in less industrialized parts of the world with lesser socioeconomic development. Nonetheless, global travel, emigration, and dietary habits utilizing foods from exotic cuisines can transplant helminth infections to nonendemic geographies [3].

Helminths can colonize the host for a long time (decades) with or without causing symptoms. Some helminth infections can cause lethal and devastating complications, such as cholangiocarcinoma caused by *Clonorchis sinensis* and portal hypertension caused by *Schistosoma mansoni*. By contrast, ingestion of ova of some other helminths, such as low-grade Necator americanus (hookworm) infection in the rural United States [4], only causes a self-limited colonization with no overt pathology.

Helminth infections trigger an immune response in their mammalian hosts but are often able to regulate that response to promote the survival of the parasite and suppress expulsion, although a comprehensive understanding of the mechanism of the latter is lacking [5]. Helminthic regulation or immunity is clearly beneficial for the parasite, but as we will see throughout this review, the induction of immune regulatory pathways can have favorable effects on the host and the host’s immune function (Figure 1 and Figure 2). We will evaluate the latter in the context of the hygiene hypothesis

## 2. Hygiene Hypothesis

Investigating more than 17,000 children who were born one week of March 1958 and followed for 23 years, the British epidemiologist David Strachan observed that the frequency of hay fever was lower in families with multiple children [30]. He proposed that the increased personal cleanliness and reduced incidence of cross infection among households with fewer people was a reason for the increased frequency of hay fever. Increased exposure to diverse infectious agents or commensals in larger families (e.g., “farm living”) appears to protect individuals from asthma or reactive airway disease based on epidemiological studies in industrialized societies [31]. These observations have expanded to other autoimmune disorders, where exposure to various bacteria, viruses, or parasites has been shown to exert a protective effect on the development of autoimmunity or immune disorders [32].

Accordingly, the rapid increase in the prevalence of immune disorders in the second half of the 20th century can be attributed to the dramatic decrease in the diversity of infectious exposures in the same time period. This decrease in diversity is associated with high-quality hygienic practices, widespread use of vaccines, application of public health measures, and use of antibacterial medications [33]. A north–south gradient in the frequency of immune disorders, where the industrialized and hygienic north displays a higher frequency of immune disorders and a reduced frequency of infectious diseases—and the south shows an opposite pattern—further supports the hygiene hypothesis as an environmental factor predisposing to immune disorders [34].

The north–south gradient also applies to helminth infections, which are frequent in the south and infrequent in the north where winter freezes kill larval forms. This pattern has been demonstrated for various infectious agents, including *Strongyloides stercoralis* and hookworms (*Necator americanus* and *Ancylostoma duodenale*). Likewise, immune disorders appear to inversely correlate with helminth infections in endemic geographies [29]. The same north–south gradient has also been proposed regarding the predisposition to inflammatory bowel diseases (IBD) [33]. 

## 3. Etiopathogenesis of Inflammatory Bowel Diseases (IBD)

IBD is a chronic inflammatory disorder of the gastrointestinal tract that is caused by genetic susceptibility and environmental factors [35,36,37]. It is generally categorized into two subtypes, Crohn’s disease (CD) and ulcerative colitis (UC), based on clinical pathological features. More than 200 genetic loci have been identified to be associated with or to influence the risk of IBD, and more than 10% of IBD patients have a family history. Nonetheless, CD has only 50% concordance between monozygotic twins and this number is even lower in UC [38]. This indicates that environmental factors and lifestyle exposures are also important in the pathogenesis of IBD. Examples of such environmental factors and lifestyle exposures include cigarette smoke, body mass index, and nutrient exposures, such as omega-3 and fatty acid levels [39]. Medications, such as NSAIDs or intestinal infections, play a role in causing acute flares. In addition, alterations in domestic lifestyle, such as the use of refrigeration have also been implicated in the development of IBD [40]. In the context of lifestyle changes in the second half of the 20th century, hygienic lifestyles have been proposed to cause IBD as well.

## 4. Hygiene Hypothesis, Inflammatory Bowel Disease, and Helminths

The results of epidemiological studies support the hygiene hypothesis for the predisposition to develop IBD: In one population-based study, several of Dr. Strachan’s original observations in hay fever [30] were reproduced in patients with Crohn’s disease [41]. This study showed that Crohn’s disease is more common in families with small numbers of children and siblings of higher birth order are more frequently affected in larger families. The same study also found that individuals living in an urban environment were more prone to suffer from IBD [41]. For people living in rural areas, the lower risk to develop IBD was later confirmed. The effect was significant for UC and CD and more dramatically pronounced in children and adolescents [42]. Dietary habits, lifestyle factors, or environmental exposures, which include helminths, were proposed as protective environmental factors. Other systematic literature reviews have suggested protective effects of *Helicobacter pylori* exposure, helminths, breastfeeding, growing in families with multiple children, and being a younger child in altering the propensity to develop IBD and protecting from the immune disorder [43].

The hygiene hypothesis is part of a larger framework linking environmental factors, such as urbanized, industrialized lifestyles, to a rise in allergic and autoimmune diseases, such as IBD. Recent large-scale prospective cohort studies have emphasized the role of processed food, including soft drinks, refined sweetened foods, salty snacks, and processed meat, in elevating the risk for developing IBD [44,45]. The epithelial barrier hypothesis links these observations with the hygiene hypothesis by proposing that inflammation in the epithelium and barrier damage, as seen in IBD, is caused in part by barrier-damaging agents, including those found in processed food, which promote opportunistic pathogen colonization, microbial dysbiosis, and thereby disorders of immune dysfunction [46].

The validity of the association between childhood exposure to environmental factors and the propensity to develop IBD via microbial dysbiosis was confirmed by studies on antibiotic use. Retrospective analyses have revealed an association between early childhood antibiotic use and IBD development [47,48]. This latter finding was recently verified by a case-control study [49]. Antibiotics alter the composition of intestinal microbiota, which appears critical for a healthy immune system development in early life according to recent and well-designed mechanistic studies [50,51,52]. The human or mammalian intestine is estimated to harbor more than 10^13^ bacteria belonging to more than 2000 species from different phyla. A balanced presence of gut bacterial species promotes health, whereas an imbalanced presence can promote the emergence of inflammatory conditions such as Crohn’s disease [53,54]. This is called microbial dysbiosis. IBD-associated dysbiosis results from the loss of beneficial strains, such as Bacteroides or Firmicutes [54], and enrichment for inflammation-driving Enterobacteriaceae [55]. Enterobacteriaceae appear enriched in the colon of IBD patients that are in remission, which suggests that IBD dysbiosis is critical to the chronicity of inflammation [56]. In this IBD-microbiota-dysbiosis framework, helminths receive further attention because they can alter the composition of microbiota in the gut (Figure 1), increase strain diversity in the cecum [57], and regulate IBD in various animal models [22] where gut microbiota (or helminthic alteration thereof) can be a critical component of regulation [58].

De-worming trials have been recommended by the World Health Organization (WHO) in regions of the world where helminths are endemic with the intention of reducing the incidence and prevalence of helminths infections as well as the potentially lethal and devastating complications associated with helminth infection (Table 1). However, these trials have revealed an increased likelihood to develop various autoimmune and metabolic diseases with the use of anti-helminthic drugs [24,59,60,61]. This puts into question the ambitious missions of the WHO and others to “de-worm the third world” [62] and highlights the concept of evolutionary mismatch. In accordance with the hygiene hypothesis, evolutionary mismatches explain that certain traits which were once beneficial for survival become maladaptive in an altered environment. In the case of the aforementioned de-worming trials, the use of antihelminthics may predispose to aberrant immune reactivity (Table 1) in an analogous fashion to predisposition to immune disorders after excessive antibiotic use during childhood [49].

Taken together and based on evidence, the hygiene hypothesis proposes that a progressively hygienic lifestyle in industrialized societies has removed us from our natural surroundings (with commensal or pathogenic microorganisms including helminths) which drove adaptation and further maturation of our immune system [63]. In the context of rural living and the hygiene hypothesis, the impact of exposure to natural surroundings on the protection from immune diseases is most pronounced in children [42]. Indeed, unique alterations to microbiota early in life through breastfeeding and weaning results in activation of regulatory T cells (Tregs) that appear critical for protection from immune diseases in animal models [50,51,52]. Helminths also stimulate Tregs [6,13,16,64] (Table 2). Breastfeeding and exposure to helminths can also protect from immune disorders based on epidemiological and clinical studies [43]. With evidence that helminthic regulation of IBD can depend on microbiota [58], studies on helminth-associated microbiota [1] and future research on helminthic immune regulation can uncover novel immune regulatory pathways important for protection from or management of IBD as well as other immune diseases.

**Table 1 biomedicines-11-01200-t001:** Adverse and beneficial effects of helminth infections.

Complications of Infections with Pathological Helminths	Complications in the Absence of Helminth Colonization (Commensal or Pathological)
• Iron deficiency anemia [65]	• Allergen skin sensitization [24,29,66]
• Vitamin B12 deficiency [67,68]	• Predisposal to IBD [43,63]
• Loeffler syndrome [69]	• Susceptibility to other immune-mediated diseases [22,66,70]
• Biliary and bowel obstruction [69]	• Predisposition to metabolic syndrome [59,60]
• Cancer ((cholangiocarcinoma) [71,72]; carcinoma of the bladder [73])	• Predisposition to type 2 diabetes mellitus [74]
• Hepatic fibrosis and portal hypertension [75]	
• Other non-communible diseases [76]	

## 5. Animal Models of Helminth Infection

Intestinal colonization with helminths triggers a type 2 immune response in animal models [77,78] and patients [79]. In this framework, animal models of helminth infections have played a critical role in understanding immune regulation in IBD in addition to other immune disorders (Table 2). Various helminths have been used in animal models to explore the immune responses they trigger in mammalian hosts and to better characterize the mechanism of helminth-induced regulation of immune disorders. These parasitic or commensal worms include *Schistosoma mansoni*, *Hymenolepis diminuta*, *Ascaris suum*, *Heligmosomoides polygyrus bakeri*, *Litomosoides sigmodontis*, *Nippostrongyloides brasieliensis,* and *Trichinella spiralis* [80]. Among these helminths, the murine nematode *Heligmosomoides polygyrus bakeri* (*Hpb*) has been a treasured tool to characterize helminth-induced immune pathways and to explore helminthic regulation of immune disorders in mouse models [81]. *Hpb* infection mimics *Necator americanus* infection in humans, which still occurs in rural United States [4]. *Hpb* infection has been used as an immune modulator to treat IBD or celiac disease, as we will see below. Based on these observations, *Hpb* infection in mouse models has enabled researchers to investigate the hygiene hypothesis in a setting that provides a cause–effect link between the immune regulated state in the host (i.e., protection from or regulation of an immune disorder) with responsible mechanisms (e.g., helminth infection and subsequent induction of immune regulatory pathways).

**Table 2 biomedicines-11-01200-t002:** Mouse models of IBD that utilized the nematode *Hpb*.

Cellular Protein or Cytokine	Associated Effector T Cell Response	*H. polygyrus bakeri (Hpb)* in Murine Colitis or GVHD Model	Publication(s)	Murine (Colitis) Model(s)
IFNγ	Th1	↓	[6,7,8,10]	GVHD PAC IL10−/− RAG−/− TCT * TNBS Wild type
IL12	Th1	↓	[10]	PAC IL10−/−
IL17	Th17	↓	[9,12]	RAG−/− TCT Wild type
IL12 (p40)		↓	[7,12]	TNBS Wild type
IL4	Th2	↑	[6,7,8,9]	GVHD TNBS Wild type
IL5	Th2	↑	[7,8,9]	TNBS Wild type
IL10	Th2, Treg	↑	[7,8,9,12]	TNBS RAG−/− TCT Wild type
IL13	Th2	↑	[7,10]	PAC IL10−/− TNBS
TGFβ	Th17/Treg	↑	[8]	Wild type
Smad7	TGFβ-antagonist	↓	[82]	Wild type
CTLA4+	Treg	↑	[12]	RAG−/− TCT
Foxp3+	Treg	↑	[10,12,13]	PAC IL10−/− RAG−/− TCT Wild type
TNFα	Th1	↓	[6,13]	GVHD Wild type
Total IgE	Th2	↑	[83]	Food allergy
Antigen-specific IgE	Th2	↓	[83,84]	Food allergy Reactive airway disease

* RAG−/− TCT. RAG−/− mice reconstituted with CD25-depleted T cells.

With the development of bacteriology and a better understanding of the link between pathogens and infections, two bacteriologists, Robert Koch and Friedrich Loeffler, proposed Koch’s postulates in the late 19th century to prove that a microorganism causes an infectious disease (Table 3) [85]. In an analogous fashion, the colonization of mice with *Hpb* or other helminths has enabled scientists to prove that a pathogen or commensal (in our case helminths) is the cause of immune regulation rather than the cause of a disease. Studies on *Hpb*-mediated immune regulation have lent solid support to epidemiological data which have demonstrated that a hygienic lifestyle (thus lack of exposure to commensals or pathogens) predisposes to immune diseases.

*Hpb* and other helminths have been used in numerous murine models of IBD to investigate immune regulation (Table 2) [86,87]. Some of these models use mucosal toxins to create injury and thereby trigger intestinal inflammation. These models are based on observations in humans that medications such as nonsteroidal anti-inflammatory drugs (NSAIDs) can cause mucosal injury, cause intestinal disease that mimics IBD [88], or cause an IBD flare [89,90]. Among these models are the dextran sodium sulfate (DSS)-, dinitrobenzene sulfonic acid (DNBS)-, or trinitrobenzene sulfonic acid (TNBS)-induced colitis models, which trigger colitis by damaging the gut–epithelial barrier, thus activating the host immune system against autologous proteins. The piroxicam-accelerated colitis in interleukin-10-deficient (IL10−/−) mice is another model where piroxicam (an NSAID) uniformly initiates chronic colitis in IL10−/− mice through epithelial barrier disruption and penetration of luminal bacteria [91]. This piroxicam-accelerated colitis (PAC) model in IL-10−/− mice mimics several features of Crohn’s disease [92]. The severity of colitis in all murine models of colitis can be scored by physical factors such as weight loss and colon length, in addition to a histological score characterized by infiltration of inflammatory cells into the colon, ulceration, and crypt abscesses [12].

Using *Hpb* in the PAC IL-10−/− model, our laboratory initially demonstrated that colonization with helminths inhibits colitis and reverses intestinal pathology while suppressing the Th1-associated proinflammatory cytokines, interferon-γ (IFNγ) and interleukin 12 (IL12), and upregulating the Th2 cytokine IL13 and the expression regulatory T cell (Treg)-specific transcription factor Foxp3 in small bowel lamina propria mononuclear cells (LPMC) [10]. Further studies have shown that *Hpb* colonization protects mice from TNBS-induced colitis while upregulating the Th2-associated cytokines, IL4, IL5, and IL13, and the regulatory cytokine, IL10 [7] (Table 2). Using the PAC IL10−/− model again, helminth-induced suppression of both pro-inflammatory IFNγ and IL17 production was shown to depend on helminth-induced production of IL4 and IL10 [9].

Similar studies from other groups using *Hpb* have further deciphered its immunoregulatory mechanisms, revealing an important role for macrophages, dendritic cells, and increased levels of opioids, such as MOR1, POMC, and β-endorphin, in attenuating TNBS, DNBS, or DSS-induced colitis [20,21,22,25,26,86,93]. Besides *Hpb*, colonization of murine colitis models with other helminths, such as Trematoda and Cestode species, also suppresses colitis through similar mechanisms [86]. Infection by the nematode, *T. spiralis,* for example, attenuates DNBS-induced colitis by downregulating Th1-type cytokines [14], while exposure to eggs of the trematode *S. mansoni* protects mice from TNBS-induced colitis by diminishing IFNγ levels and enhancing IL-4 production [11]. Exposure to *S. japonicum* eggs also prevents TNBS-induced colitis by increasing the Treg immune response compared with the Th17 immune response and altering the metabolism of infected mice, namely, inhibiting the glycolysis and lipogenesis pathways while promoting fatty acid oxidation [15].

The role of helminthic immune regulation of IBD has also been assessed in immunological models of colitis that do not require administration of a mucosal toxin, such as DSS or TNBS, where the intestinal injury is rather mediated by administration of naïve T cells in the absence of T regulatory cells. The best characterized of these models is the colitis model caused by naïve T cell transfer into lymphopenic RAG−/− mice where in vivo activation of transferred T cells during homeostatic expansion triggers colitis (transfer colitis) [94]. Transfer colitis studies have demonstrated that helminths can employ the innate immune system [25] and specifically the dendritic cells [26] to regulate colitis. Within the adaptive arm of the immune response, transfer colitis studies have also attested to the importance of colonic Foxp3+ Tregs [95] and to novel regulatory CD8 T cells [96] in helminthic regulation of IBD in mice. Helminth colonization of T- and B-cell deficient RAG mice has also shown an important role of innate lymphoid cells in helminth eradication [27].

Another model of immunological colitis does not require T-cell transfer. Mice with T-cell-specific deficiency of TGFβ signaling develop spontaneous colitis [97], in which we have demonstrated that helminth-induced regulation of colitis and generation of regulatory IL10 requires intact TGFβ circuitries [8,98]. Further studies using a model of graft-versus-host disease (GVHD) and GVHD-associated colitis, a disease that mimics IBD, have shown that helminths augment TGFβ generation by inducing the Th2 pathway [13,64]. In the context of TGFβ, studies have also shown that helminths generate products that can regulate aberrant immune reactivity by stimulating host TGFβ receptors [16]. Current studies are attempting to identify helminth products that can be used to prevent or treat immune diseases [99], including IBD in animal models [100]. In addition to generating products that can activate regulatory immunity by triggering the host TGFβ pathway, helminths have also been shown to decrease expression of the TGFβ inhibitor Smad7 by host cells, and using a T-cell transfer model of colitis, the critical role of Smad7 downregulation in helminthic regulation of IBD was shown [82].

Studies have also shown that lymphoid and nonlymphoid memory T cells maintain the Th2 and immunoregulatory response originally elicited by helminth infection even after clearance of helminths [101]. These memory T cells include CD8 virtual memory lymphocytes [102]. Although a possible negative impact of helminth infection on memory response after vaccination has been a concern [103,104,105], colonization with helminths can reduce the severity of infections with other pathogens such as COVID-19 [106]. Moreover, the helminth-induced Th2 pathway can promote memory T-cell-mediated control of viral infection [107].

## 6. Clinical Studies with Helminths in Patients with Inflammatory Bowel Diseases

In the context of the hygiene hypothesis, authorities believe that childhood exposure to helminths can contribute to protection from immune disorders in adult life. Demonstration that *Hpb* infection in mice can regulate established chronic colitis [10] gave rise to helminth trials in IBD patients. Phase 1 clinical trials for UC and CD have used two nematode species—the pig whipworm *Trichuris suis*, an intestinal pathogen of pigs that can briefly colonize humans [108], and *Necator americanus*, a hematophagous hookworm. *T. suis* ova (TSO) where chosen because the helminth eggs can be isolated in a specific-pathogen manner.

The safety of TSO in IBD patients was first evaluated in an open trial in a small cohort of 4 patients with active CD and 3 patients with UC [109], and again later in 12 patients in a randomized, double-blinded, placebo-controlled, sequential-dose escalation trial [110]. In this latter trial, varying doses of TSO were administered to patients with CD, where the patients received one dose of either 500, 2500, or 7500 viable embryonated TSO or placebo. Similar to the earlier safety study [109], TSO was well tolerated in all doses, including the highest, and none of them caused any significant adverse effects [110].

Several studies have shown a clinical benefit in using TSO to treat IBD patients. In the aforementioned open-label trial at the University of Iowa, 7 IBD patients were given a dose of 2500 *T. suis* ova (TSO) resulting in remission for 6 of the 7 patients according to the IBD Quality of Life Index. Even if the beneficial effects in this study were only temporary, maintenance therapy with 2500 TSO every three weeks sustained clinical improvement without causing adverse effects [109]. A subsequent randomized, double-blinded, placebo-controlled trial treated 54 active UC patients with 2500 TSO or placebo orally at 2-week intervals for 12 weeks [111]. At 12 weeks, statistically significant differences between the TSO- and placebo-treated patients were present, where 13 of 30 patients (43.3%) with ova treatment demonstrated improvement according to the intent-to-treat principle as determined by the UC Activity Index compared with 4 of 24 patients (16.7%) given placebo.

A similar trial for active CD enrolled 29 patients, yielding a decreased CD Activity Index (CDAI) of nearly 79.3% and a remission rate of 72.4% in patients treated with TSO for 24 weeks [112]. This trial was open label, however; therefore, a high placebo effect could not be ruled out. Indeed, a larger, randomized, double-blinded, placebo-controlled trial in Europe, where 252 adults with active CD received 6 total doses of 250, 2500, and 7500 TSO or placebo every 2 weeks, revealed an unexpectedly elevated clinical remission rate in the placebo recipients (42.9%) compared with the TSO recipients (38.5%, 35.2%, and 47.2%) [113]. Despite these drawbacks, TSO was shown to promote a dose-dependent immune modulatory effect.

In clinical trials with helminths, *N. americanus* poses an alternative to *T. suis* as, unlike *T. suis, N. americanus* has a frequency of causing mild parasitic infections in humans and can survive for several years within the body [114]. Infective larvae (L3i) are acquired percutaneously and adult *N. americanus* worms reside in the small intestine. An initial proof-of-concept study in CD patients revealed a decrease in clinical disease activity after the administration of L3i [115]. After 20 weeks, 8 of the 9 patients treated with L3i demonstrated signs of remission with a decreased CDAI. Even if this study yielded noticeable side effects associated with helminth use, including pruritis and a hookworm-related enteropathy, the side effects were matchlessly less severe compared with the side effects of immune modulators or biologics, which constitute the standard of care in moderate to severe IBD and can cause lethal infections and predispose to cancer or demyelinating diseases, besides other things. Therefore, the interest in using helminths as a safe alternative to the current state-of-the-art therapy has continued. In this context, helminth products have also been investigated. P28 S-glutathione transferase (P28SGT), a protein derivative from a trematode parasitic helminth, appeared safe when injected into CD patients in a multicenter, open-label, pilot Phase 2a study, and decreased disease activity index and inflammatory marker scores [116].

Celiac disease (CeD), like IBD, is an immune-mediated disease manifesting with damage to the gastrointestinal tract and chronic intestinal inflammation [117]. Unlike IBD, however, the principal triggering antigen of CeD is well known (gluten), and the removal of dietary gluten can achieve remission. Therefore, the host–parasite interaction can be examined independently of potential artifacts in CeD, making it uniquely suited to study the relationship between helminth infection and treatment of intestinal inflammation [118]. In a 21-week double-blinded, placebo-controlled pilot study, 20 CeD patients inoculated with L3i or placebo and subjected to a 5-day oral wheat tolerance challenge displayed immune responses, with decreased inflammatory cytokines and a weak trend towards attenuation of a response to gluten [118,119]. In accordance with the previous IBD trial [115], adult *N. americanus* hookworms in the intestine showed mild but noticeable side effects.

A subsequent 52-week trial involving 12 CeD patients inoculated with 20 L3i and subjected to escalating gluten challenges revealed decreased proinflammatory interferon-γ (IFNγ) and increased Treg cells in addition to improved symptoms and quality of life scores in the infected CeD patients; yet histopathological scores remained insignificantly altered [120]. This study was small (*n* = 12) and not placebo controlled. Nonetheless, a larger placebo-controlled trial conducted more recently confirmed the findings of its predecessor by demonstrating that *N. americanus* infection strikingly reduced the tissue transglutaminase (tTG) titer, even if it did not improve tolerance to moderate gluten consumption (2g/d) but still improved symptoms and quality of life [121]. Although hookworm infection does not cure celiac disease, understanding the improved symptoms and immunological and serological profiles in these studies is clinically relevant for understanding the role of hookworm infection in other diseases, including IBD. A microbial analysis of CeD patients infected with *N. americanus* suggests a potential mechanism whereby hookworms could increase tolerance to gluten consumption by maintaining the composition of the intestinal microbiota [122].

Like CeD, multiple sclerosis (MS) is also relevant within the context of IBD as the two diseases share epidemiologic features and the severity of MS is inversely correlated with concurrent helminth infection [123]. Previous clinical trials of MS patients treated with TSO demonstrated increased Th2-associated cytokines [124], increased Th2 cytokine interleukin-4 (IL4) producing and decreased IL2 producing T cells, mild elevation of peripheral eosinophil counts [125], and decreased active brain lesions [126]. However, a study concluded no clinical efficacy despite helminth-induced eosinophilia [127]. A recent prospective, randomized, placebo-controlled, double-blind, phase II trial confirmed the variable clinical efficacy [128]. In this study, 5 MS patients received 2500 TSO orally every 2 weeks for 12 months. Increased HLA-DR+CD4+ T cells in treated patients was apparent, suggesting a helminth-specific adaptive immune response, but there was variability in Treg cell frequencies and T-cell responses across individuals. Other studies have confirmed that despite the lack of statistical significance in decreasing active MS brain lesions, TSO and *N. americanus* treatments have been deemed safe, immunobiological effects with increased Tregs have been observed, and potentially favorable MRI outcomes have been observed [129,130]. Again, maintenance of microbial diversity may play a significant role in the attenuation of disease severity as MS patients treated with *N. americanus* revealed stable alpha diversity and expanded *Parabacteroides* with 16S rRNA high-throughput sequencing [131]. 

In addition to IBD, CeD, and MS, clinical trials for helminth therapy have been conducted for numerous immune-mediated diseases, such as allergic rhinoconjunctivitis, autoimmune encephalitis, peanut allergy, asthma, plaque psoriasis, rheumatoid arthritis, and type-1 diabetes [132,133,134,135], in addition to metabolic disorders, including type-2 diabetes and abdominal obesity [74,136]. All of these studies have demonstrated varying levels of success, but supervised exposure to helminthic therapy has always been safe with at worst minor side effects.

Helminth-derived immunomodulatory products, such as P28SGT, among many others, show great promise for future clinical trials in the treatment of IBD and other immune-mediated diseases [99]. For example, a structurally novel TGF-β mimic, Hp-TGM, interacts with mammalian TGFβ receptors and upregulates Tregs [100], and excretory/secretory products (ES) from the gastrointestinal helminth *Nippostrongylus brasiliensis* modulate the immune response associated with type-2 diabetes [137].

A genetic approach for future treatments is also promising. Genome-wide association studies have revealed that IBD is a group of polygenic disorders in which hundreds of loci are implicated [38]. Among these loci are components of the human leukocyte antigen (HLA) complex [138], in which genes are directly linked to increased pathogen diversity. Perhaps this knowledge can improve stratification within clinical research studies whereby the genotypes associated with IBD can be analyzed and stratified for genes that have evolved due to pathogenic pressures (not excluding those within the HLA complex), thus allowing clinical researchers to identify cohorts with said genotypes. A similar study design has already been proposed for clinical trials using helminths in individuals with Alzheimer’s disease who are ApoE 4/4 carriers, which is associated with pathogen richness [139].

## 7. Conclusions and Future Directions

Treatment of moderate to severe IBD involves the use of immune modulators and/or biologics, which have several toxicities and side effects, such as predisposition to infections, cancer, and demyelinating diseases in addition to others. Helminths or helminth products are attractive therapeutic avenues given their safety profile. Their beneficial effects have been reported, although larger clinical trials are likely needed to better characterize any role in the care of IBD patients. Epidemiological research on helminths together with basic science studies on helminthic immune regulation or the effects of helminth products on immune regulation have attested to common regulatory pathways critical to immune regulation in the setting of the hygiene hypothesis, oral tolerance [140], or helminthic immune modulation, such as TGFβ or helminth product TGFβ-mimics [141,142,143] and altered microbiota [144]. Further research on helminths is expected to be rewarding in physiological and biochemical characterization of not only helminthic immune modulation to identify novel, potent, and safe approaches to treat IBD but also to result in discovery of novel and common immune regulatory pathways useful in the management of other immune pathologies.

## Figures and Tables

**Figure 1 biomedicines-11-01200-f001:**
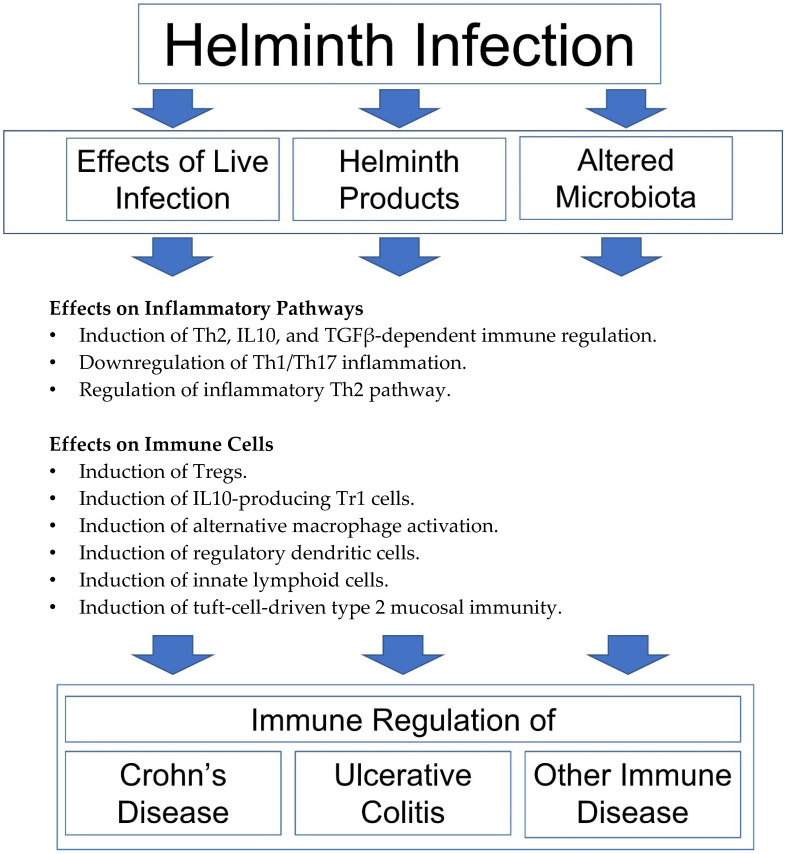
Mechanisms of helminthic immune regulation. Helminth-modulated immunity results in the induction of Th2, IL10, and TGFβ-dependent immune regulation [6,7,8,9,10,11], downregulation of Th1/Th17 inflammation [6,7,8,9,10,11,12,13,14,15], regulation of inflammatory Th2 pathway [16,17], induction of Foxp3+ Tregs [10,13,15,18], IL10-producing Tr1 cells [19], alternative macrophage activation [20,21,22,23,24], regulatory dendritic cells [25,26], innate lymphoid cells [27] and tuft-cell-driven type 2 mucosal immunity [23,28,29].

**Figure 2 biomedicines-11-01200-f002:**
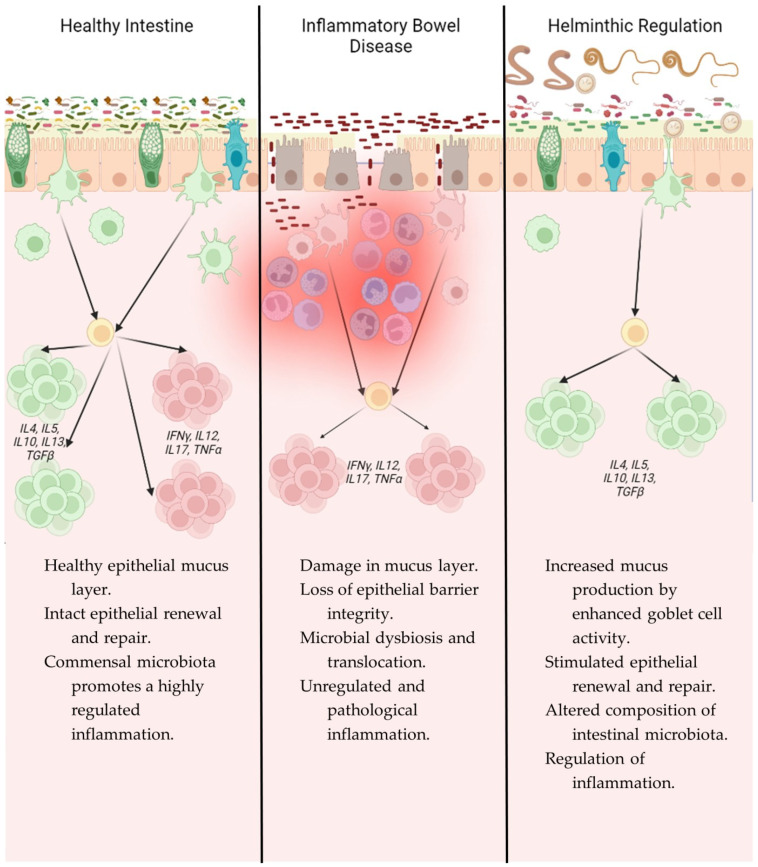
Immunological comparison of intestinal mucosa in a healthy state, IBD, and after helminth infection.

**Table 3 biomedicines-11-01200-t003:** Koch’s postulates.

No.	Koch’s Postulates
1	The microorganism must always be demonstrated in all diseased cases.
2	The microorganism must be isolated and grown in culture.
3	The microorganism grown in culture must cause disease when inoculated into a healthy and susceptible laboratory animal.
4	The microorganism must be re-isolated from the experimentally infected subject.

## Data Availability

No new data is generated for this review article. Please refer to the corresponding author of the citations for inquiries related to original data or archived datasets.

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
