# Peer review of "Helminth Lessons in Inflammatory Bowel Diseases (IBD)"

_biomedicines, 2023, doi:10.3390/biomedicines11041200_

Round 1

Reviewer 1 Report

Tyler et al. extensively illustrated and presented the relevant literature about IBD and the symbiotic, parasitic, and therapeutic potential of helminths in inflammatory diseases with a specific focus on IBD. However, there are some necessary quires that should be addressed and presented before consideration for publication.

Major comments

1) Figure 1, where the authors illustrate about the helminth infection (page-2), couldn't properly demonstrate all the information. As it is the review article, a more comprehensive diagrammatical representation is needed.

2) As per the title of the review article "The helminths lessons", it could be better to draw the role of helminths in gut homeostasis in the diagram as mentioned helminth infection (figure 1). Moreover, more detail is needed about helminths and IBD as the title covers a wide range of information about the problem.

3) Why the authors specifically focused on immune disorders with a particular focus on IBD, as per the literature, there are many different factors (environmental, host-physiology, host genetics, dietary lifestyle). Could the authors kindly search for relevant literature about all these factors influencing the onset of IBD or idiopathic inflammatory diseases with respect to helminths' symbiotic, parasitic, and therapeutic potential?

Other comments

1) On page 2, in the hygiene hypothesis (3rd to 10th sentence), the paragraph is confusing and couldn't properly explain the scenario and thus creating ambiguity. On one side it is mentioned that "personnel cleanness and reduced incidence of cross infection among the household was the reason for increased frequency of hay fever" and those protect individuals from asthma etc, while on other hand, it shows a protective effect on the development of autoimmunity, or immune disorders. So, it could be better to simplify the structure of the sentences and present them in a more comprehensive form.   2) On page 2, in the hygiene hypothesis (sentence 13th), the word "Culprit" is not looking suitable from a scientific point of view. So, the sentence should be rephrased. 

Author Response

Reviewer 1:

  • Quote: Tyler et al.extensively illustrated and presented the relevant literature about IBD and the symbiotic, parasitic, and therapeutic potential of helminths in inflammatory diseases with a specific focus on IBD. However, there are some necessary quires that should be addressed and presented before consideration for publication.
    • Response: We thank the reviewer for the critique. Please find our responses below.
  • Quote: 1) Figure 1, where the authors illustrate about the helminth infection (page-2), couldn't properly demonstrate all the information. As it is the review article, a more comprehensive diagrammatical representation is needed.
    • Response: We extended Figure 1 and added another figure (Figure 2) to give further details about helminthic immune modulation. We detailed cell types and regulatory pathways associated with helminthic immune modulation. We also included references.
  • Quote: 2) As per the title of the review article "The helminths lessons", it could be better to draw the role of helminths in gut homeostasis in the diagram as mentioned helminth infection (figure 1). Moreover, more detail is needed about helminths and IBD as the title covers a wide range of information about the problem.
    • Response: We hope that the new Figure 1 and Figure 2 display a comprehensive review of helminth-induced intestinal immune regulation and physiological alterations in the gut in healthy state and IBD.
  • Quote: 3)Why the authors specifically focused on immune disorders with a particular focus on IBD, as per the literature, there are many different factors (environmental, host-physiology, host genetics, dietary lifestyle). Could the authors kindly search for relevant literature about all these factors influencing the onset of IBD or idiopathic inflammatory diseases with respect to helminths' symbiotic, parasitic, and therapeutic potential?
    • Response: We added a section (section 3) to answer this query. The title of the section reads as “3. Etiopathogenesis of inflammatory bowel diseases” (page 4).

Other Comments (Reviewer 1)

  • Quote: 1) On page 2, in the hygiene hypothesis (3rd to 10th sentence), the paragraph is confusing and couldn't properly explain the scenario and thus creating ambiguity. On one side it is mentioned that "personnel cleanness and reduced incidence of cross infection among the household was the reason for increased frequency of hay fever" and those protect individuals from asthma etc, while on other hand, it shows a protective effect on the development of autoimmunity, or immune disorders. So, it could be better to simplify the structure of the sentences and present them in a more comprehensive form.  
    • Response: We thank the reviewer for careful review. We reworded the section to avoid said ambiguity. This paragraph reads as “Investigating more than 17,000 …” (page 4).
  • Quote: 2)On page 2, in the hygiene hypothesis (sentence 13th), the word "Culprit" is not looking suitable from a scientific point of view. So, the sentence should be rephrased. 
    • Response: We reworded this paragraph. This paragraph reads as “Accordingly, the rapid increase…” (page 4).

Reviewer 2 Report

Manuscript No Biomedicines-2267173

Helminth Lessons in Inflammatory Bowel Diseases (IBD)” for Biomedicines

1.      I am asking the authors to analyse and systematize the possible benefits and losses resulting from the use of anti-helminthic drugs and compare them to the benefits and losses of the body resulting from the development of specific pathogens and helminths.

2.      I also ask you to analyse the presented concepts in the context of immunological memory. Is colonization with helminths that may inhibit colitis an effective solution in relation to accidental infection, fighting it and acquiring immunity resulting from the appearance of memory cells.

3.      How should colonization by helminths be interpreted in the context of inflammation development. Can inflammation be interpreted only as an inducer of an immune response or maybe an adverse effect that may contribute to the weakening of the body?

4.      In addition, please mention more about the development of innate immunity in infections by helminths. Will the potential increase in IgE levels not have adverse health consequences?

Author Response

Reviewer 2

  • Quote: I am asking the authors to analyse and systematize the possible benefits and losses resulting from the use of anti-helminthic drugs and compare them to the benefits and losses of the body resulting from the development of specific pathogens and helminths.
    • Response: We added a table (the new Table 1) to the manuscript to address this query in the paragraph under section 4. This paragraph reads as “De-worming trials have been recommended …” (page 5). We also added references to the table (the new Table 1).
  • Quote: I also ask you to analyse the presented concepts in the context of immunological memory. Is colonization with helminths that may inhibit colitis an effective solution in relation to accidental infection, fighting it and acquiring immunity resulting from the appearance of memory cells.
    • Response: We thank the reviewer for this comment. We added a paragraph to section 5 (the last paragraph of this section), which reads as “Studies have also shown that lymphoid and nonlymphoid …” (page 10). In this paragraph, we attempted to summarize recent observations on the impact of helminth colonization on generation of protective memory against other infections.
  • Quote:   How should colonization by helminths be interpreted in the context of inflammation development. Can inflammation be interpreted only as an inducer of an immune response or maybe an adverse effect that may contribute to the weakening of the body?
    • Response: We addressed this query in Figure 2 where we display features of pathological inflammation in IBD and how helminth infection can regulate it. In general, uncontrolled chronic inflammation in different immune mediated diseases can weaken the body where helminthic regulation can potentially interfere with this adverse outcome.
  • Quote:    In addition, please mention more about the development of innate immunity in infections by helminths. Will the potential increase in IgE levels not have adverse health consequences?
    • Response: We included various elements of the innate immune system that contribute to helminthic immune regulation in new Figure 1 with references. We also added the regulatory effect of helminth infection on antigen specific IgE production to Table 2. As our manuscript focuses on IBD, we have preferred not to add a paragraph discussing the effects of helminths on allergic and IgE responses.

Round 2

Reviewer 1 Report

The authors addressed all the comments raised and can be acceptable for publication.